# Numerical and Experimental Investigation of the Effect of Traffic Load on the Mechanical Characteristics of HDPE Double-Wall Corrugated Pipe

**Hongyuan Fang** [1,2,3,4]**, Peiling Tan** [1,2,3]**, Xueming Du** [1,2,3,]*****, Bin Li** [1,2,3]**, Kangjian Yang** [1,2,3] **and Yunhui Zhang** [1,2,3]

1   School of Water Conservancy Engineering, Zhengzhou University, Zhengzhou 450001, China; 18337192244@163.com (H.F.); tpl1752311002@163.com (P.T.); 13523519067@163.com (B.L.); 13253671350@163.com (K.Y.); 15950597992@163.com (Y.Z.)
2   National Local Joint Engineering Laboratory of Major Infrastructure Testing and Rehabilitation Technology, Zhengzhou 450001, China
3   Collaborative Innovation Center of Water Conservancy and Transportation Infrastructure Safety, Zhengzhou 450001, China
4   Southern Engineering Inspection and Restoration Technology Research Institute, Huizhou 516029, China
*   Correspondence: 2007-dxm@163.com; Tel.: +86-17737348392

**Abstract:** The high-density polyethylene (HDPE) double-wall corrugated pipe, which is a kind of flexible pipe, is widely used in municipal drainage networks. The characteristics of the surrounding soil and pipe bed, pipe cover depth, backfill compaction, type of pavement and pavement design, and traffic loads are some of the major factors that affect the stress and deformation of pipes. In this study, the ABAQUS 3D finite element model was used to analyze the influence of backfill compactness, traffic loads, diameter, and hoop stiffness on the mechanical characteristic of an HDPE pipe under traffic loads. A series of full-scale tests were carried out to verify the validity of the simulation results. For the conditions tested, the results showed the following: (1) the Von-Mises stress of the pipe was mainly determined by the earth pressure at the crown, and the stress caused by backfill compaction increased significantly but had a short duration and limited impact on the pipe; (2) traffic load alone had little influence on the mechanical behavior of the pipe: while under the action of the loose backfill in contact with the pipe, the pipes were more sensitive to the traffic load response; (3) the fluctuations in the Von-Mises stress of the pipe mainly depended on the magnitude and speed of the traffic load; (4) for pipes with a small diameter, non-compacted backfill easily caused stress concentration in the pipe, while the degree of backfill compaction had almost the same effect on the distribution of stress for pipes with different hoop stiffness.

**Keywords:** HDPE pipe; backfill compactness; Von-Mises stress; traffic load

## 1. Introduction

The high-density polyethylene (HDPE) corrugated pipe is a type of flexible pipe that has played a significant role in municipal engineering. The earth's pressure of buried flexible pipes is shared between the pipe and soil. Since the stiffness of the pipes is smaller than that of the surrounding soil, when a load is applied to the soil surface, the relative downward deflection of the adjacent soil prism is less than that of the central soil prism, the central soil prism is subjected to an upward friction force (Figure 1), and the earth pressure acting on the pipe is less than the weight of the central soil prism [1].

Therefore, the structural behavior of a buried flexible pipe is influenced by the hoop stiffness of the pipe and the stiffness of the backfill [2]. The stiffness of the backfill depends mainly on the degree of compaction [3], water content [4], soil properties [5], and grade [6]. Therefore, the influence of the earth pressure above a flexible pipe on pipe deformation cannot be ignored, and the load acting on pipes can be divided into soil loads and external loads. At the same time, under different soil conditions, the force transmitted from external load to the crown of the pipe is different [7,8]. In many previous studies, scholars have mainly analyzed the mechanical deformation law of flexible pipes by laboratory tests and numerical simulations.

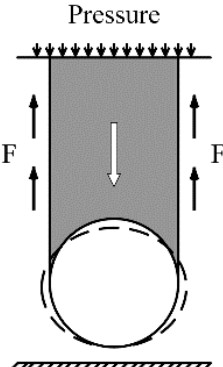

**Figure 1.** Mechanism of soil arching within the soil-pipe system.

During the installation of a pipe, its deformation is affected by the compaction method [9], compaction equipment [10], groove width, backfill material, the thickness of the backfill pipe cover, pipe diameter, and pipe hoop stiffness [11]. Masada and Sargand (2017) studied the deformation of a flexible pipe during its installation and proposed a formula for calculating the peak deflections of a thermoplastic pipe by backfilling the soil to the crown of the pipe [12]. On the basis of field test data and numerical simulation data, Zhou et al. proposed a calculation formula for the deflection of an HDPE pipe during backfilling [11]. Through field tests, You et al. measured the deflection and hoop strain generated during the pipe backfilling process, and they proposed an empirical relationship to predict the hoop strains that were generated on HDPE pipes in the construction phase [13]. For a flexible pipe buried in backfill with asymmetrical compactness, weak support on the springline leads to an increase in the lateral displacement of the pipe; hence, the haunch exhibits strain concentration behavior [14]. Talesnick et al. (2011) designed a new technique to measure the radial contact pressure applied to a buried pipe and the deflection of the pipe under static vertical loading, and the measurement results indicated that the radial contact pressure and deflection of the pipe were sensitive to the compaction backfill [15]. Terzi et al. (2012), through a series laboratory tests on HDPE pipes buried in poorly graded sand with different compaction, found that the stress and deformation of these pipes were higher when the surrounding backfill was loose, and the maximum stress appeared at the invert, whereas the crown stress and invert stress were higher when the backfill was compact [16]. In addition, the method of installation and the type of backfill material were also key factors that affected the deformation of the pipe. A laboratory experiment was carried out by Brachman et al. (2008) to compare the effects of compaction methods, gradation, and soil material on the strain within profiled thermoplastic pipes; the empirical strain coefficient was derived by using measured data, which simplified the calculation of the maximum local bending strain in a special-shaped pipe design method [9].

With the popularization of finite element analysis software, numerical simulation has been widely applied to the calculation of pipe deformation. Dhar et al. (2004) used a two-dimensional (2D) model to calculate deflections and circumferential strains of a profiled thermoplastic pipe and examined the effectiveness of the model by a biaxial pipe test [17]. The simulation results revealed that the pipe stress was concentrated near the loose backfill area when loose backfill was placed under the haunch. Using ABAQUS, which a finite simulation model software by the SIMULIA Company in

France, Abri and Mohamedzein (2010) established a 2D model to evaluate the deformation of HDPE and polyvinyl chloride (PVC) pipes installed in dune sand; the effects of several factors (such as relative density, backfill material, and soil cover) on the pipe performance were analyzed, and the results showed that the response of pipes increased with the increase in the relative density of the sand [18]. A three-dimensional (3D) finite element model for analyzing the deformation of pipes under normal fault motions was proposed by Naeini et al. (2016), and the influences of pipe diameter, thickness, buried depth, friction angle, and the density of the backfill on the bending strain of the pipe were discussed. The simulation results showed that the deformation of the pipe was sensitive to the buried depth and backfill type [19]. Kejie et al. (2019) established a 3D model by ABAQUS to investigate the mechanical characteristic before and after repairing the pipe with carbon fiber reinforced polymer (CFRP) layer [20].

With the development of roads and transportation, traffic load has become one of the critical factors of pipe deformation. Peindl et al. (1992) conducted a similar field study on buried PVC pipes and demonstrated that a PVC pipe buried in flowable fly-ash backfill was feasible and safe under continuous high-load applications [21]. At present, when dealing with the influence of traffic loads on pipes, the traffic load is often simplified to the static load without considering the pavement structure, which leads to pipe deformation that is inconsistent with the real traffic load. Many researchers believe that simplifying the traffic load to the static load results in increased pipe deformation and, therefore, safer conclusions. However, Hyodo and Yasuhara (2017) carried out low-embankment traffic load tests on clay and found that the vertical load generated by the dynamic load was more than three times that of the static load [22]. In the 1960s, the fourth power theory proposed by the American Association of Interstate Highway Workers (AASHO) indicated that the static axle load was more suitable for simulating traffic load under low-speed and light axle load conditions [23]. For concrete pipe, the results of finite element simulation showed that pipe deformation caused by a vehicle load could be ignored when the buried depth was 1.5 times greater than the diameter of the pipe [24]. In generally, the influence of traffic load on concrete and other rigid pipelines cannot be ignored [25–27]. At the same time, a well-compacted backfill would also reduce the influence of the external load on the pipe. Mohamedzein and Aghbari (2016) found that plastic pipes were less affected and less elastically deformed by external loads when the backfill was compacted. In addition, the stress and deformation of the pipe were susceptible to the traffic load because of the shallow buried depth of the pipe [28]. Zhan and Rajani (1997) simplified the traffic load to the uniform static load in a nonlinear, two-dimensional finite element model and revealed that low-strength backfill material could significantly reduce the impact of the traffic load on PVC pipes [29].

As can be seen from the above literature review, the deformation of buried plastic pipes under traffic loads has been studied by using field tests or two-dimensional finite element models. However, there is little information about the mechanical response of HDPE pipes under a traffic load and local loose backfill soil. In this work, an ABAQUS 3D finite element model of an HDPE pipe was established to study the effects of parameters, such as soil compaction during the installation, the buried depth of pipe, the compactness of the backfill surrounding the pipe, traffic load, pipe diameter, and pipe hoop stiffness, on the Von-Mises stress of the pipe. Then, a series of full-scale tests were carried out to verify the validity of the simulation results.

## 2. Finite Element Formulations

### 2.1. Model of the HDPE Pipes

Because of the complexity of the crest and valley geometry of an HDPE pipe, the soil grid was irregular when establishing the model, which made it challenging to simultaneously increase the calculation accuracy and efficiency of the model in the actual structure (Figure 2). Therefore, it was necessary to simplify the pipe into a three-dimensional straight-walled pipe.

As a profiled pipe, the double-wall corrugated pipe is carefully designed with a particular cross-sectional shape of the corrugated exterior wall and straight interior, so its weight is much lower than that of the straight-walled pipe and other profiled pipes with the same hoop stiffness [30]. Under the same installation conditions, the interaction mechanisms between the pipe and soil are similar between straight-walled pipes and double-wall corrugated pipes with the same hoop stiffness [20,31]. A straight-walled pipe with the same hoop stiffness can be used to replace a profiled pipe in the 2D finite model since field tests have proved the model to be suitable for the deflection and strain of the double-wall corrugated pipe [21,32,33]. In this study, the straight-walled pipe was used to simulate the corrugated pipe through the hoop stiffness transformation method, which could better simulate the mechanical response of the pipe under external loads. From the hoop stiffness calculated in Equation (1), the stiffness of a straight-walled pipe is obtained from Equation (2):

$$S_P = EI/D^3 \tag{1}$$

$$S_P = Et^3/12D^3 \tag{2}$$

where $S_P$ is the hoop stiffness of a flexible pipe, $E$ is the elastic modulus of the pipe, $I$ is the pipe wall moment of inertia per unit length of the longitudinal section, $t$ is the equivalent wall thickness, and $D$ is the nominal diameter of the pipe.

In the simplified double-wall corrugated pipe, the diameter of the pipe remains unchanged, and the wall thickness of the straight-walled pipe is calculated using the hoop stiffness. The corresponding parameters of the simplified double-wall corrugated pipe, i.e., the inner diameter, the wall thickness, and the hoop stiffness, are summarized in Table 1. Although high-density polyethylene is a viscoplastic material, the deformation of the pipe shows linear elasticity in the short term [34]. In the elastic model, the axial displacement of the pipe is constrained, and eight-node reduced-integration shell elements (S8R) are used to simulate the pipes [23,35]. The material parameters of HDPE pipes suggested by the code of the Technical Specification for Buried Plastic Drainage Pipe Engineering (CJJ 143-2L10) are shown in Table 2 [36].

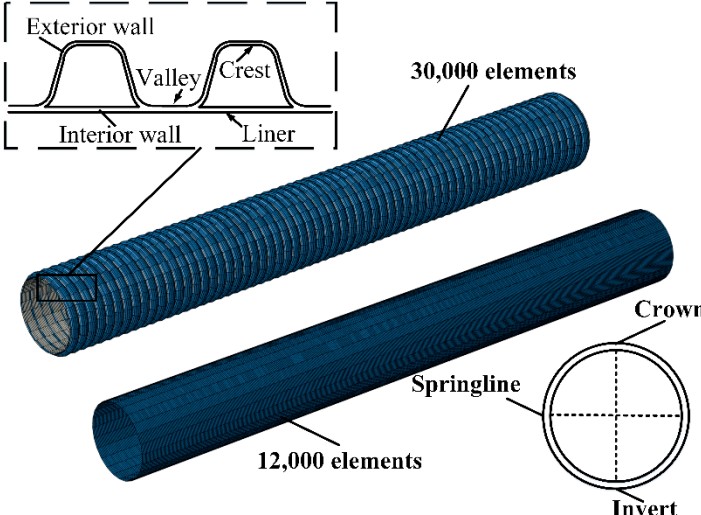

**Figure 2.** Element numbers of double-wall corrugated pipe and straight-walled pipe.

**Table 1.** Shape parameters of the simplified straight-walled pipe.

| Hoop Stiffness (kN/m$^2$) | Nominal Diameter (mm) | Wall Thickness (mm) |
|---|---|---|
| 4 | 400 | 15.7 |
| 4 | 600 | 24.0 |
| 4 | 800 | 31.3 |
| 8 | 600 | 29.6 |
| 10 | 600 | 31.9 |

**Table 2.** Physical properties of the high-density polyethylene pipe.

| | Degree (kg/m$^3$) | Elastic Modulus (MPa) | Poisson's Ratio |
|---|---|---|---|
| HDPE pipe | 950 | 800 | 0.4 |

### 2.2. Dimensions of the Road Structure

The whole 3D dataset of the road structure and the traffic load location are displayed in Figure 3. A model with dimensions of 16 × 20 × 6 m was adopted to reduce the influence of the boundary on the pipe [1]. The dynamic load was applied to a region (0.4 × 10 m) of the wheel track with a wheelbase of 1.7 m.

The road layers consisted of pavement (asphalt concrete, AC-20), base (asphalt-treated base, ATB-30), subbase (cement treated base, CTB), and subgrade (soil). In this study, the elastic constitutive model extended by a non-associated flow rule was used to describe the behavior of the pavement [37], whose parameters are presented in Table 3. The subgrade was simulated by employing the Mohr–Coulomb constitutive model [33,38], and the standard parameters specified are presented in Table 4. The road elements were replaced by the eight-node reduced-integration element (C3D8R) [39], the bottom plane of the road was fully constrained, and the horizontal displacements of four vertical planes were constrained [40]. The thickness and material of the backfill considered in the numerical simulations are shown in Figure 4.

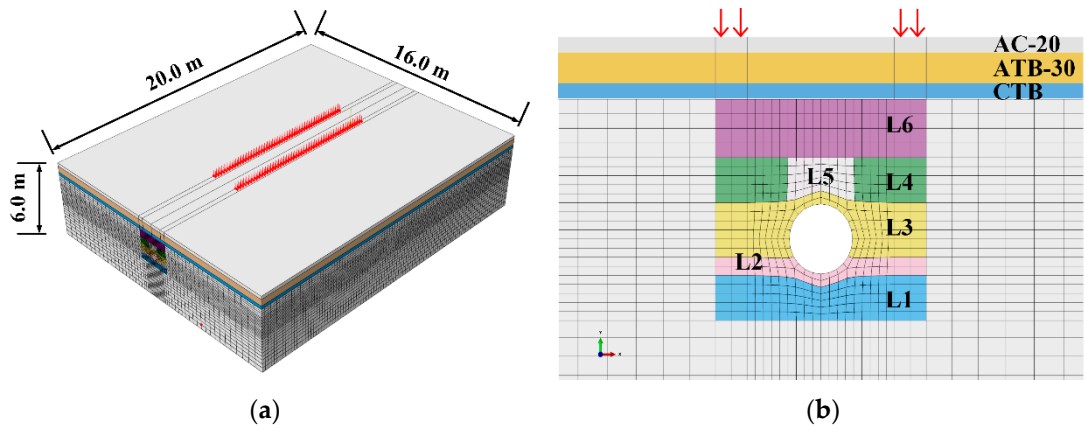

| (a) | (b) |
|---|---|

**Figure 3.** (**a**) Three-dimensional data of the road structure; (**b**) the layers of the road structure.

**Table 3.** Physical properties of the pavement in numerical analysis.

| Pavement Layer | Density (kg/m$^3$) | Elastic Modulus (MPa) | Poisson's Ratio | Thickness (m) |
|---|---|---|---|---|
| AC-20 | 1200 | 2400 | 0.35 | 0.2 |
| ATB-30 | 1000 | 2400 | 0.30 | 0.4 |
| CTB | 1500 | 2300 | 0.20 | 0.2 |

**Table 4.** Physical properties of the standard compacted backfill in numerical analysis.

| Backfill Layer | Dry Density (kg/m³) | Compaction Degree (%) | Elastic Modulus (MPa) | Angle of Internal Friction (°) | Cohesion (kPa) | Poisson's Ratio |
|---|---|---|---|---|---|---|
| L1 | 1650 | 90 | 5 | 35 | 15 | 0.25 |
| L2 | 1650 | 95 | 7 | 26 | 10 | 0.25 |
| L3 | 1650 | 95 | 7 | 26 | 10 | 0.30 |
| L4 | 1650 | 85 | 3 | 26 | 18 | 0.30 |
| L5 | 1650 | 90 | 5 | 35 | 15 | 0.30 |
| L6 | 1650 | 90 | 5 | 35 | 15 | 0.30 |
| In-situ soil | 2020 | - | 30 | 28 | 20 | 0.30 |

**Figure 4.** The thickness and material of the backfill in numerical analysis.

## 2.3. Pipe Installation Procedure

During backfill soil compaction, the soil exerts passive earth pressure on the pipe. Low-stiffness flexible pipes are prone to large deformations under lateral earth pressure, so the deformation of the pipe during installation cannot be ignored [41]. The Mohr–Coulomb criterion expresses the relationship between the soil vertical stress and the soil lateral horizontal stress:

$$\sigma_h = \sigma_v (1 + \sin\theta)/(1 - \sin\theta) + 2c\sqrt{(1 + \sin\theta)/(1 - \sin\theta)} \qquad (3)$$

where $\sigma_h$ is the horizontal stress, $\sigma_v$ is the vertical stress, $c$ is the cohesion of soil, and $\theta$ is the friction angle of soil.

The cohesion ($c$) of sand as backfill is assumed to be 0 to simplify the calculation. For purely friction materials, Taleb and Moore (1999) proposed an empirical compaction coefficient, $K_n$ [41]. Then,

$$\sigma_h = K_n \sigma_v (1 + \sin\theta)/(1 - \sin\theta) \qquad (4)$$

When a vibratory plate tamper is used to compact the soil, $K_n = 1$; when an impact tamper is used to compact the soil, $K_n = 2$ [42].

On the basis of the above theory, Elshimi and Moore (2013) established a two-dimensional pipe–soil model using ABAQUS to analyze the deformation of the pipe during the assembly process. A horizontal load was applied to the pipe joints to simulate the lateral earth pressure on the pipe during the compaction process, and the simulation results were verified by using published data to prove the accuracy of the method [43]. In this research, a similar method was employed to simulate the installation of HDPE pipes (Figure 4).

## 2.4. Traffic Load Data

In this study, a three-axle vehicle was assumed to apply the traffic load to the pipe; the front axle was a single-wheel, and the rear axle was a dual-wheel. The contact area (A) of the single-wheel with the ground was approximately regarded as consisting of a rectangle with a radius of 0.4 × 0.6 L and two semicircles with a radius of 0.3 L, and the contact area between the tire and the ground was

simplified, as shown in Figure 5 [44]. The problem was simplified by assuming that the size of the tire action area on the ground was 0.2 × 0.3 m, and the pressures on the ground were the same.

When a vehicle is moving, the contact area between the tire and the ground and the pressure will change [45]: the problem was simplified by assuming that the area of the contact between the tire and the ground remained constant, and the pressure change in dynamic vehicle load was simulated by a sinusoidal curve. When the dynamic load changes with the law of positive half-chord pulse, the simulation of different real loading modes is achieved by controlling the amplitude and period of the pulse [46]. The effect of the vehicle dynamic load on a point is given by Equation (5) [47]:

$$F(t) = p + q(t) \tag{5}$$

where $p$ is the static pressure of the wheels, $q(t)$ is the attached dynamic load, and $t$ is time.

The attached dynamic load is equivalent to the sinusoidal distributed load [48]:

$$q(t) = q_{max}sin^2(\pi/2 + \pi t/T) \tag{6}$$

$$T = 12L/v \tag{7}$$

where $q_{max}$ is the magnitude of the vertical wheel dynamic overload, $T$ is the load cycle, $L$ is the length of the tire tread imprint, and $v$ is the vehicle speed.

When the road is rough, and the vehicle speed is slow, the dynamic amplification factor is set to 1.3. Thus,

$$q_{max} = (1.3 - 1)p = 0.3p \tag{8}$$

$$F(t) = p + 0.3psin^2(\pi/2 + \pi t/T) \tag{9}$$

Because of the variation in the position and pressure of the tire with time, in the numerical simulations, loads were applied to a cell, and the movement of the load was simulated by applying loads to different cells. If the distance between the front axle and rear axle is assumed to be 2 m, then the center point of the line connecting the front axle and rear axles is the center of gravity of the load, and the distance moved by the center of gravity is the distance traveled by the dynamic load. A fixed time analysis step was used in the iterative calculation to ensure the accuracy of the calculation results. When each iteration was completed, the load moved forward by one unit. Initially, the rear tires were located at a and b, and the front tires were located at c; after two iterations, the rear tires were located at a' and b', and the front tires were located at c'. The movement process of the vehicle load is shown in Figure 6. As the load moved forward along the cell, the magnitude of the load fluctuated in the form of a sinusoidal curve and circulated until the application of the load stopped.

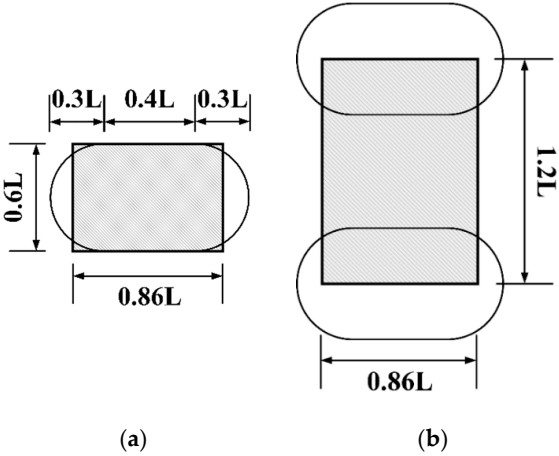

(a)　　　　　　　　　　　(b)

**Figure 5.** *Cont.*

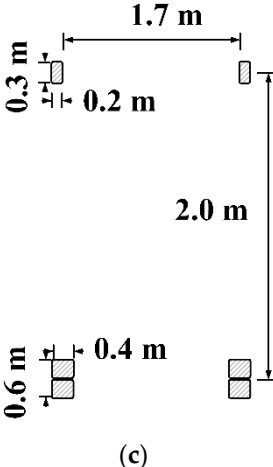

**(c)**

**Figure 5.** Simplified tire grounding area: (**a**) single-wheel; (**b**) double-wheel; (**c**) a ten wheels' truck.

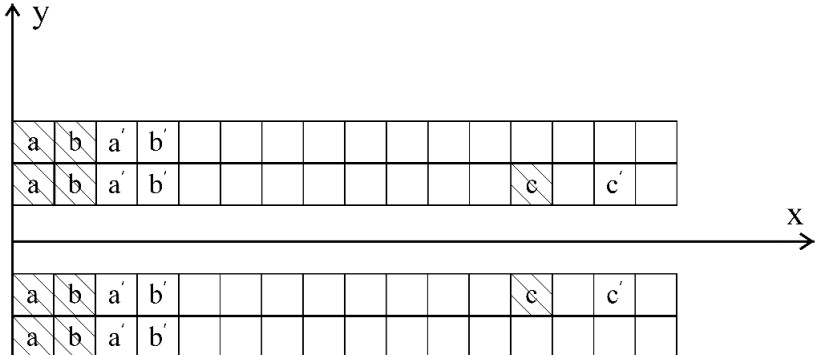

**Figure 6.** Moving load random loading process.

*2.5. Simulated Conditions*

The base and soil around the pipe support it and restrain the deformation of the pipe. As a result of the limitation of compaction methods, it is difficult for the backfill from the invert to the springline of the pipe to reach the required compaction degree, whereas the backfill above the pipe is easily compacted, and loose backfill is less common for engineering above the pipe. The compactness of the bedding layer (L1) and that of the side-fill layers (L2 and L3) has a measurable effect on the stress state and deformation of the flexible pipe [32]. This paper focused on the impact of the backfill compactness of L1, L2, and L3 on the stress of the pipe.

The mechanical parameters of the soil vary with the compaction degree, and the elastic modulus has the most significant influence on the interaction between the soil and the pipe. In this part of the study, a decrease in the compaction degree was simulated by reducing the elastic modulus of the backfill. The relationship between soil compactness and the elastic modulus is presented in Table 5 and based on the structural design code of pipes of water supply and wastewater engineering (GB 50332-2002) [49]; i.e., the backfill material was gravel, its compactness was 90%, and the backfill modulus was 5 MPa. The standard compaction degree of L1 was set to 90%, and that of L2 and L3 was 95%; the compaction degrees of these layers were reduced to 85% and defined as loose (the limit state of the backfill was not compacted) in order to simulate weak compaction. When the compaction degree of the backfill was 85% and loose, the elastic modulus was 3 and 1 MPa, respectively.

The backfill compaction state was taken as a series, and the magnitude and speed of the traffic load and the diameter and hoop stiffness of the HDPE pipe were changed under various compaction conditions. The simulation conditions are listed in Tables 6 and 7.

**Table 5.** Relationship between the compaction degree and the elastic modulus of the backfill.

| Backfill Material | Elastic Modulus E (MPa) | | | |
| --- | --- | --- | --- | --- |
| | Compaction Degree (%) | | | |
| | 85 | 90 | 95 | 100 |
| Gravel | 5 | 7 | 10 | 20 |
| Gravel, sand pebbles, and fine-grained soil content less than 12% | 3 | 5 | 7 | 14 |
| Gravel, sand pebbles, and fine-grained soil content greater than 12% | 1 | 3 | 5 | 10 |
| Clay, silt, and sand content greater than 25% | 1 | 3 | 5 | 10 |
| Clay, silt, and sand content less than 25% | — | 1 | 3 | 7 |

**Table 6.** Simulation backfill compaction conditions.

| Backfill Condition * | Elastic modulus (MPa) | | | | | |
| --- | --- | --- | --- | --- | --- | --- |
| | Lay 01 | Lay 02 | Lay 03 | Lay 04 | Lay 05 | Lay 06 |
| **Standard** | 5 | 7 | 7 | 3 | 5 | 5 |
| **L1-loose** | 1 | 7 | 7 | 3 | 5 | 5 |
| **L1-85%** | 3 | 7 | 7 | 3 | 5 | 5 |
| **L2-loose** | 5 | 1 | 7 | 3 | 5 | 5 |
| **L2-85%** | 5 | 3 | 7 | 3 | 5 | 5 |
| **L3-loose** | 5 | 7 | 1 | 3 | 5 | 5 |
| **L3-85%** | 5 | 7 | 3 | 3 | 5 | 5 |

* In the numerical simulation, the magnitude and speed of traffic load, the hoop stiffness and diameter of the pipe are changed respectively under each backfill condition, as shown in Table 7.

**Table 7.** Simulation traffic load and pipe physical variables.

| Load Magnitude (MPa) | Speed (km/h) | Diameter (mm) | Hoop Stiffness (kPa) |
| --- | --- | --- | --- |
| 0.5<br>0.7<br>1 | 60 | 600 | 4 |
| 0.7 | 30<br>40<br>50 | 600 | 4 |
| 0.7 | 60 | 400<br>800 | 4 |
| 0.7 | 60 | 600 | 8<br>10 |

## 3. Simulation Results and Sensitivity Analysis

### 3.1. General

The most common failure mode of buried flexible pipes is a ductile failure because the pipes creep after being subjected to a long-term stable load [50]. The von-Mises criterion states that when the equivalent stress of a certain stress state reaches a certain value related to the stress state, the material yields, and the Von-Mises stress is this equivalent stress. Von-Mises stress can be expressed by the stress component, as shown in Equation (8). Previous studies have focused on the circumferential stress and changes in the diameter of the flexible pipe to indicate the deformation of the pipe [9,16,23,51]. Compared with the real stress, the von-Mises stress comprehensively considers the deformation of materials in multiple directions, and the assessment of the ductile failure of pipes is more accurate; thus, it is reasonable to take the Von-Mises stress as a parameter that indicates the deformation of the

pipe [52–54]. When the corrugated pipe is simplified as a straight-walled pipe, the circumferential and axial stress distribution of a particular section of the corrugated pipe cannot be obtained, so the Von-Mises stress is more suitable than the real stress to define the deformation of the pipe. In the tests described in the following subsections, the Von-Mises stress was calculated and output by ABAQUS, which could be used to predict the dangerous location at which the pipe is most likely to generate a ductile failure under the combined action of loose backfill and traffic loads. The Von-Mises stress in the section located at the half-length of the pipe was selected for analysis to eliminate the influence of the boundary on the calculation result Von-Mises.

$$\sigma_{Mises} = \sqrt[2]{0.5\Big[\big(\sigma_x - \sigma_y\big)^2 + (\sigma_x - \sigma_z)^2 + \big(\sigma_z - \sigma_y\big)^2 + 6\big(\tau_{xy}^2 + \tau_{zy}^2 + \tau_{xz}^2\big)\Big]} \tag{10}$$

where $\sigma_x$ is the X-axis stress, $\sigma_y$ is the Y-axis stress, $\sigma_z$ is the Z-axis stress, $\sigma_r$ is the circumferential stress, $\sigma_x$ is the axial stress, $\tau_{xy}$ is the shear stress in the y-direction on the x-plane, $\tau_{zy}$ is the shear stress in the y-direction on the z-plane, and $\tau_{xz}$ is the shear stress in the z-direction on the x-plane.

### 3.2. Influence of Backfill Compaction on the Pipe During Installation

In the process of simulating the backfill of the pipe, it was assumed that the compaction of the backfill met the requirements of the specification, and the deformation of the pipe was only affected by the compaction of the backfill. The pipe was 600 mm and 4 kPa, and the buried depth was 1.5 m. The deformation and von-Mises stress distribution of the pipes are shown in Figures 7 and 8. In Figure 7, "Initial" represents the shape of the pipe before deformation due to soil compaction, "0.15 m" represents the deformed shape of the pipe when the backfill thickness (from the invert to the surface of the backfill) was 0.15 m, and "Stable" represents the deformed shape of the pipe when it reached a stable state after installation was completed.

During the installation of the pipe, the compaction of the backfill on the pipe side caused horizontal compression of the pipe, and the deformation of the pipe was a "vertical elliptical deformation" (Figure 7). Similar conclusions have been reached by previous authors [55,56]. The vertical deflection of the pipe first increased and then decreased. When the backfill was compacted to 3/4 of the diameter of the pipe, the vertical deflection of the pipe reached its maximum value, and the deformation of the pipe was still a vertical ellipsis after the assembly was completed. This might be because when the soil was not backfilled to the springline, the pipe was only affected by the lateral earth pressure; with the increase in backfill thickness, the lateral earth pressure acting on the pipe increased. After backfilling to the springline, the pipe was subjected to the combined effect of vertical earth pressure and lateral earth pressure, and the vertical deflection of the pipe gradually decreased. As the burial time increased, the pipe changed from a vertical elliptical deformation to a horizontal elliptical deformation. The vertical elliptical deformation that occurs during the installation of a flexible pipe will reduce the deflection deformation under the action of soil covering the top of the pipe, thus prolonging the service time of the pipes [12].

Similar to the vertical deflection of the pipe, with the increase in the thickness of compacted backfill, the Von-Mises stresses of the crown, springline, and invert first increased and then decreased (Figure 8). When the vertical deflection of the pipe was at its maximum, the Von-Mises stresses also reached the maximum. When the pipe assembly was completed, the Von-Mises stress of the invert was the largest, and the Von-Mises stress of the springline was the smallest. After the pipe was buried for some time, the stress of the crown decreased slightly, the stress of the springline increased, and the stress of the invert decreased. This proved that the soil and pipe reached a new equilibrium. In the process of backfilling, the maximum Von-Mises stress of the pipe was several times that of the stable state (2.5 times at the crown, 3.2 times at the springline, and 3.4 times at the invert). However, this high-stress state of the pipe had a very short duration: in general, the pipe would not be damaged during the backfilling. Thus, although the pipe experienced great strain during installation, the

long-term load (such as the earth pressure, surface load, and traffic load) that the pipe bears was considered to have a greater impact on its deformation during its service period.

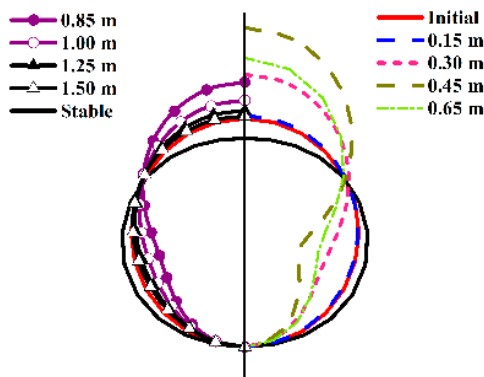

**Figure 7.** Deformed shape of the pipe during the installation.

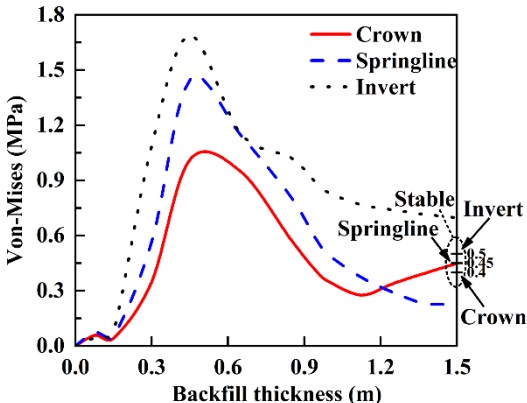

**Figure 8.** The Von-Mises stress of the pipe during the installation.

### 3.3. Influence of Buried Depth on the Pipe Under Traffic Load

When analyzing the impact of the burial depth on the pipe, the backfill soil was considered to have good compaction. The change in the buried depth of the pipe was realized by controlling the thickness of Lay 06, and the thickness of the pavement structure and that of other backfilling layers were kept constant. The magnitude and speed of the traffic load were set to 0.7 MPa and 60 km/h, and the diameter and ring stiffness of the pipe were 600 mm and 4 kPa, respectively.

As shown in Figure 9, the Von-Mises stress distribution laws of pipes with different buried depths were similar; the earth pressure played a decisive role in the Von-Mises stress value, and the stress slightly fluctuated when the traffic load was applied. The stress reached the maximum when the traffic load moved to the middle of the pipe. This result was consistent with the literature [57]. The increase in soil cover depth was important for reducing the traffic load [24], but it also caused the pipe to bear greater earth pressure. Therefore, the specification required the soil cover depth of a buried plastic pipe to be 0.5–1.0 m [36], which decreased the combined effect of traffic loads and earth pressure on the pipe.

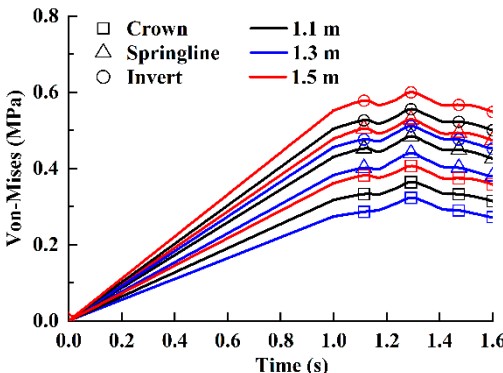

**Figure 9.** Time history curves of the Von-Mises stresses of the different buried depth of pipe under traffic load.

### 3.4. Influence of Backfill Compactness on the Pipe Under Traffic Load

In these simulations, the magnitude and speed of the traffic load were set to 0.7 MPa and 60 km/h, respectively, as base criteria. The diameter and the hoop stiffness of the pipe were 600 mm and 4 kPa, respectively. The time history curves of the Von-Mises stresses of the crown, the springline, and the invert are given in Figure 10. When the compaction degree of the pipe bed and lateral backfill decreased, the Von-Mises stress of the pipe was less sensitive to the traffic load than to the earth pressure. Figure 10a illustrates that the crown Von-Mises stress decreased as the compaction degree in L1/L2 decreased or as the compaction degree in L3 increased. The time history curve of the springline Von-Mises stress was similar to that of the crown, but the springline stress value was higher than that of the crown (Figure 10b). In addition, the invert Von-Mises stress decreased significantly when the filling media at L1 was loose, while the invert stress increased as the compaction degree of L2 decreased or the compaction degree of L3 increased (Figure 10c).

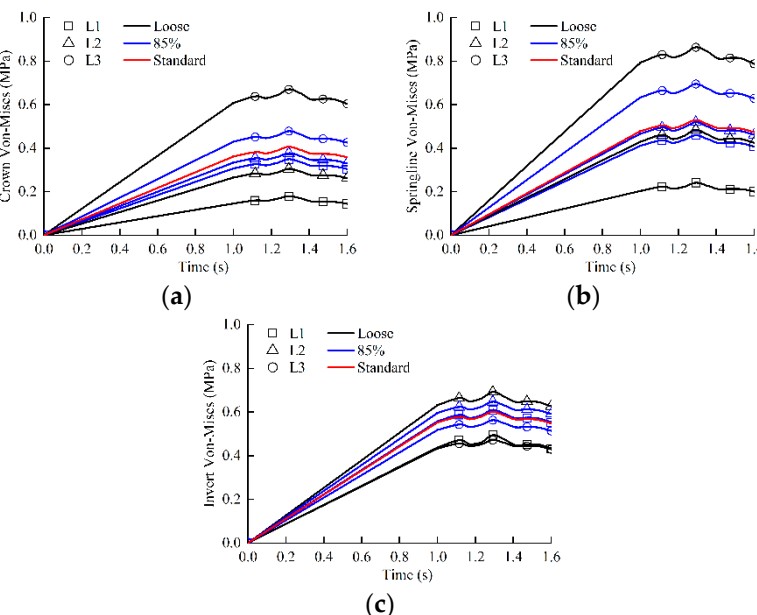

**Figure 10.** Time history curves of Von-Mises stresses under traffic load: (**a**) the crown; (**b**) the springline; (**c**) the invert.

The reason for the above results might be that the energy generated by the earth pressure was absorbed because of the looseness of the base (L1); therefore, the earth pressure on the pipe was smaller, and the Von-Mises stress decreased. However, this does not mean that the loose base is

beneficial for the operation of the pipe. Poorly compacted bases have negative effects on flexible pipes in many aspects, such as bending [57], differential settlements [58], failures in the pipe joints, and, consequently, poor hydraulic performance [59]. These aspects lead to the leakage of sewage, which pollutes groundwater, shortens the service life of the pipe, and increases the operating costs. Therefore, the compaction degree of the base is fundamental in both the design phase and in the operating conditions of any sewer network and should be highlighted.

*3.5. Influences of Traffic Loads on the Von-Mises Stress of the Pipe*

From the above analysis, the region and the compaction degree of the backfill are considered to jointly determine the Von-Mises stress of the pipe. Vehicles transmit traffic loads to pipes through the pavement structure and backfill, so the mechanical response of pipes to the traffic load is constrained by the backfill state. In the experiment described in this section, the Von-Mises stress reaction of the pipe was studied by changing the magnitude and speed of the traffic load.

The value of the stress caused by the traffic load was proportional to the load magnitude (Figure 11), and the stress fluctuated from smooth to acute with the increasing speed of the traffic load; however, the maximum and minimum stress values were almost unchanged (Figure 12). Although the compaction degree of the backfill had a limited influence on the stress caused by the traffic load, a lower compaction degree was associated with a greater fluctuation in stress. The compaction degree of L1 or L2 mainly affected the stress at the invert, and L3 had the greatest effect on the stress at the crown and springline. In general, the higher the Von-Mises stress caused by earth pressure, the greater the fluctuation in the stress caused by traffic loads. However, when L1 was loose, the invert Von-Mises stress was lowered, but the fluctuation in the stress caused by the traffic load was more substantial (Figures 11a and 12a). In general, if the backfill does not meet the required compaction degree, then the pipe region in contact with it is more sensitive to the traffic load.

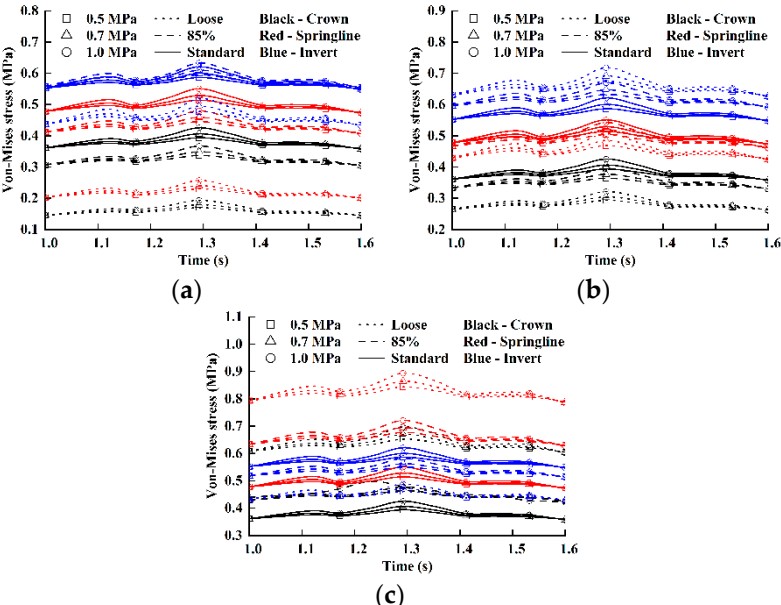

**Figure 11.** The Von-Mises stress under different load magnitudes: (**a**) change in L1 compaction degree; (**b**) change in L2 compaction degree; (**c**) change in L3 compaction degree.

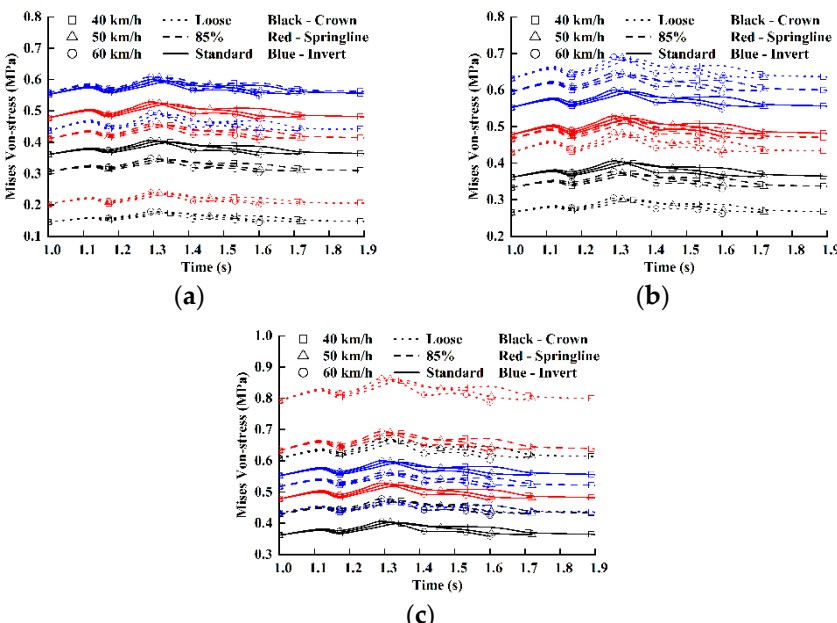

**Figure 12.** The Von-Mises stress under different load speeds: (**a**) change in L1 compaction degree; (**b**) change in L2 compaction degree; (**c**) change in L3 compaction degree.

### 3.6. Influences of Diameter on the Von-Mises Stress of the Pipe

In general, the maximum stress and deformation of PVC pipes increase as the diameter increases [60]; therefore, it can be inferred that HDPE pipes with different diameters have different responses to a traffic load under the same backfill conditions. Figures 13–15 display the Von-Mises stress curves of pipes with different diameters. In this part of the simulation, the traffic load was 0.7 MPa and 60 km/h, and the hoop stiffness of the pipe was 4 kPa.

The variations in the crown Von-Mises stress versus the diameter of pipes were similar (Figure 13). When the compaction degree of L1 was lower than the standard, the crown stresses for different pipe diameters descended in the order 400 mm pipe, 800 mm pipe, and 600 mm pipe (Figure 13a). The crown stress was inversely proportional to the diameter when the degree of compaction of L2 or L3 was reduced (Figure 13b,c). The results indicated that the crown stress of the 400 mm pipe was more sensitive to the compaction degree of L3, while those of the 600 mm and 800 mm pipes were greatly influenced by the compaction degree of L1 and L2. The response of the springline stress to the different diameters of the pipe was similar to that of the crown stress, and the stress curves of the 600 mm and 800 mm pipes were more coincident at the springline than at the crown (Figure 14). This means that when the diameter of the pipe was large enough, the deformation of the springline was almost independent of the diameter. The variations in the invert stress of the 600 mm and 800 mm pipes with backfill compactness were similar (Figure 15), but the 400 mm pipe differed at the invert, where the Von-Mises stress increased with looser L1 (Figure 15a). When the compaction degree of L1 or L2 decreased, the invert stress of the 400 mm pipe was the largest, followed by 800 and 600 mm (Figure 15a,b).

When the compaction degree of L3 decreased, the invert stresses of pipes with different diameters descended in the order 400, 600, and 800 mm (Figure 15c), which indicated that the invert stress of pipes with large diameters was more susceptible to the compaction degree of L3. In conclusion, the Von-Mises stress of the 400 mm pipe was always greater than that of 600 mm and 800 mm pipes under the same cover depth, backfilling condition, and traffic load. Therefore, it is speculated that when the diameter of the pipe is small enough, the loose backfill around the pipe and traffic load are more likely to cause pipe deformation.

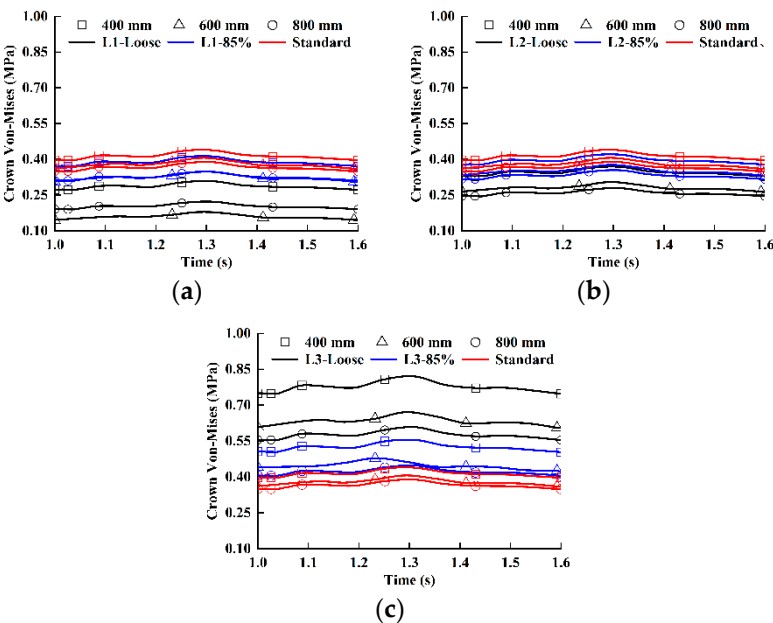

**Figure 13.** The crown Von-Mises stress of different diameter pipes under traffic load: (**a**) change in L1 compaction degree; (**b**) change in L2 compaction degree; (**c**) change in L3 compaction degree.

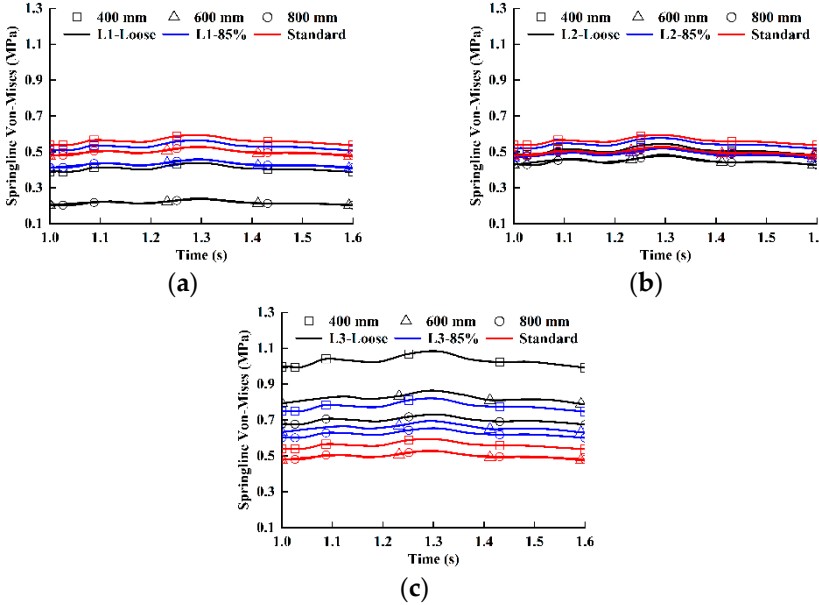

**Figure 14.** The springline Von-Mises stress of different diameter pipes under traffic load: (**a**) change in L1 compaction degree; (**b**) change in L2 compaction degree; (**c**) change in L3 compaction degree.

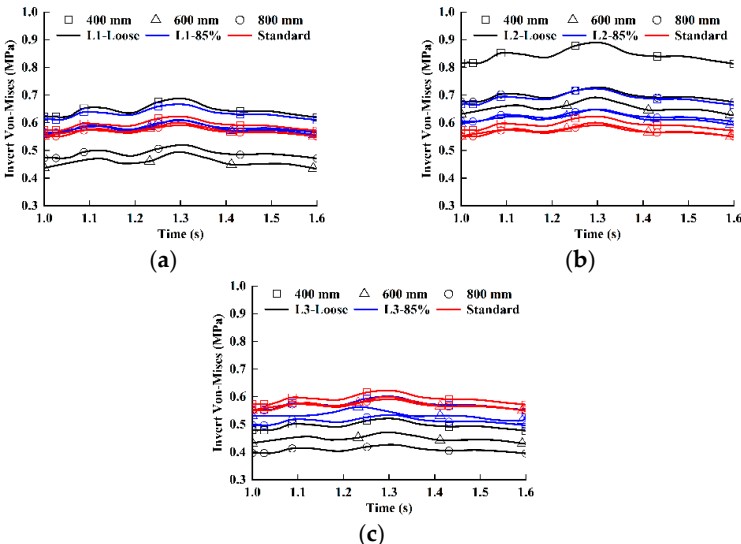

**Figure 15.** The invert Von-Mises stress of different diameter pipes under traffic load: (**a**) change in L1 compaction degree; (**b**) change in L2 compaction degree; (**c**) change in L3 compaction degree.

### 3.7. Influences of Hoop Stiffness on the Von-Mises Stress of the Pipe

The pipe-soil interaction and the ability of the pipe to resist the external force mainly depend on the hoop stiffness of the flexible pipe; therefore, the hoop stiffness of the pipe has a profound influence on the mechanical deformation of the pipe [61]. In the investigation of Von-Mises stress response of the pipe to the hoop stiffness, the values of the hoop stiffness of the 600 mm diameter pipe were varied between 4, 8, and 10 kPa, and the traffic load was 0.7 MPa and 60 km/h.

The hoop stiffness had a negligible impact on the Von-Mises stress of the pipe when the backfill was compacted according to the standard (Figure 16). This might be because an increase in hoop stiffness led to a weakening of the pipe-soil interaction and an increase in earth pressure above the crown. However, at the same time, the ability of the pipe to resist deformation was improved, and the Von-Mises stress of the pipe was almost unchanged. For clear description, the Von-Mises stress curves of pipes with different hoop stiffness were simplified into one and expressed as "Standard".

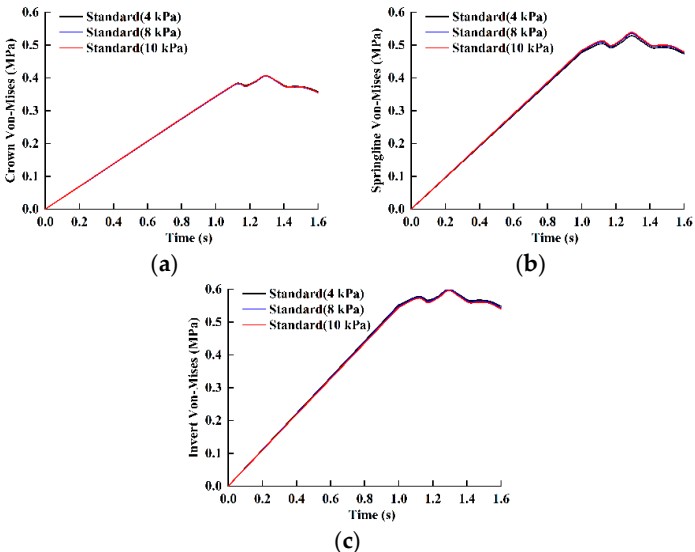

**Figure 16.** The Von-Mises stresses of different hoop stiffnesses of pipe under traffic load when the backfill compactness is standard: (**a**) the crown; (**b**) the springline; (**c**) the invert.

　　　The crown Von-Mises stress responses of pipes with different hoop stiffness to the compactness of L1 and L2 barely differed, but the response was proportional to the hoop stiffness when the L3 compactness decreased (Figure 17). The springline Von-Mises stress also had low sensitivity to the hoop stiffness, even in the case of loose backfill (Figure 18). When L2 was loose, the springline stress of the pipe with low hoop stiffness was slightly lower than that of the pipe with high hoop stiffness. At the invert, the influence of the Von-Mises stress on the hoop stiffness was evident. The worse the backfill was compacted, the greater the difference in the invert stress of the pipe for different hoop stiffness. When the compaction degree of L1 or L2 decreased, the invert stress was inversely proportional to the hoop stiffness (Figure 19a,b); when the compaction degree of L3 decreased, the invert stress was proportional to the hoop stiffness (Figure 19c). Under the same compaction conditions, the Von-Mises stresses of pipes with different hoop stiffness were not different, so the responses of these pipes to traffic loads were also very close. From these results, it could be concluded that, compared with the diameter, the hoop stiffness had a limited influence on the Von-Mises stress of the pipe.

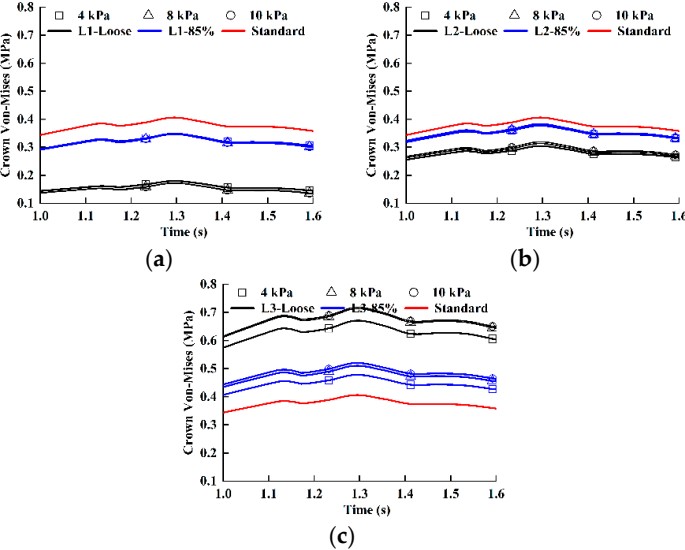

**Figure 17.** The crown Von-Mises stresses of pipes of different hoop stiffness under traffic load: (**a**) change in L1 compaction degree; (**b**) change in L2 compaction degree; (**c**) change in L3 compaction degree.

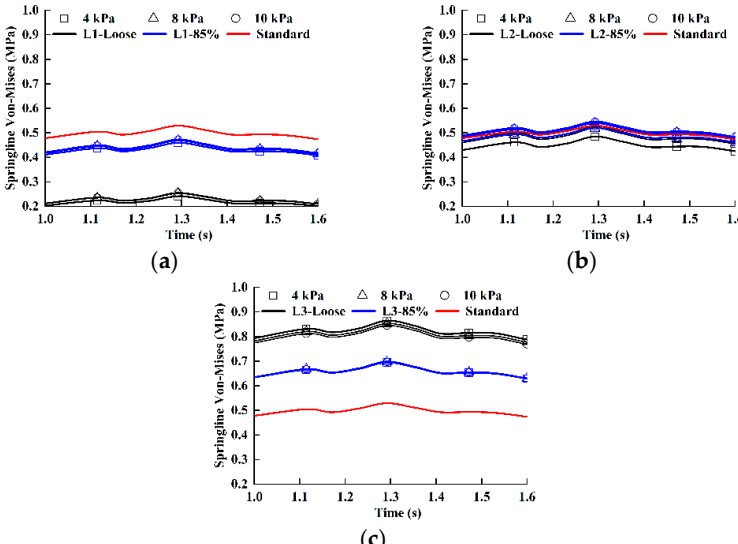

**Figure 18.** The springline Von-Mises stresses of pipes of different hoop stiffness under traffic load: (**a**) change in L1 compaction degree; (**b**) change in L2 compaction degree; (**c**) change in L3 compaction degree.

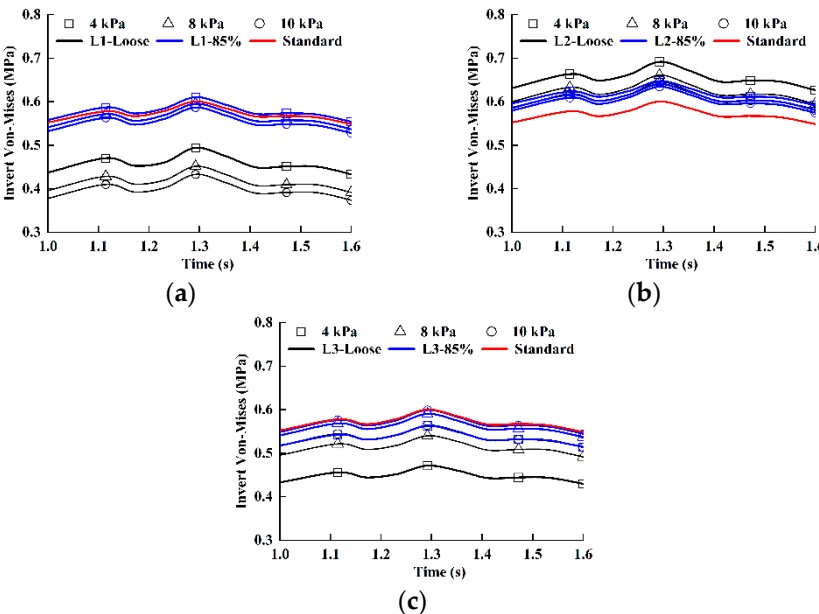

**Figure 19.** The invert Von-Mises stresses of pipes of different hoop stiffness under traffic load: (**a**) change in L1 compaction degree; (**b**) change in L2 compaction degree; (**c**) change in L3 compaction degree.

### 3.8. Influences of Backfill with Asymmetrical Compactness on the Pipe

In the field construction, it is common that backfill has different compaction degrees between the left and right regions, which leads to eccentric compression on the pipe. In this part, four conditions with different degrees of compactness on the left and right sides of the backfill were analyzed in order to evaluate the Von-Mises stress of the pipe. The Von-Mises stress caused by the earth pressure on the pipe was plotted to determine the maximum point of the Von-Mises stress, and the variation in the Von-Mises stress with the traffic load at this point was analyzed.

Figures 20–23 show that when the compaction degree of the backfill was left-right asymmetrical, the Von-Mises stress distribution was asymmetrical, and the stress was concentrated near the pipe region in contact with the loose backfill. The location and scope of the loose region jointly determined the strain distribution of the pipe, and the positional influence was more prominent (Figures 21–23). The asymmetric stress distribution was more likely to induce the failure of the whole pipe, so the long-term performance of the pipe would be adversely affected by the non-uniform compactness of the backfill. At the maximum point of the Von-Mises stress, the stress fluctuation caused by the traffic load was more significant than the stress under standard conditions, but it was still negligible. This result agreed with the findings reported by Noor and Dhar [24].

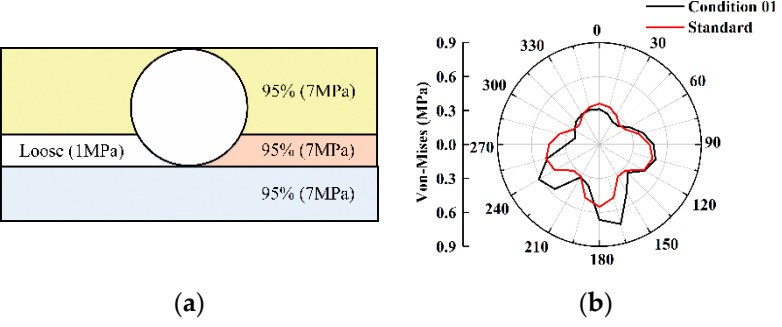

(**a**)  (**b**)

**Figure 20.** *Cont.*

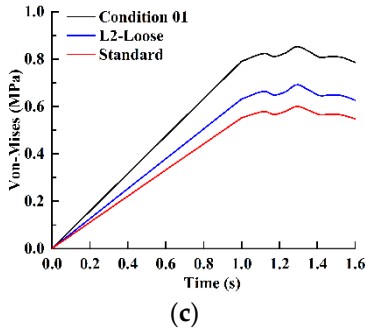

(**c**)

**Figure 20.** (**a**) The backfilling schematic of Condition 01; (**b**) the stress distribution of the pipe; (**c**) the time history curve of the right invert under Condition 01.

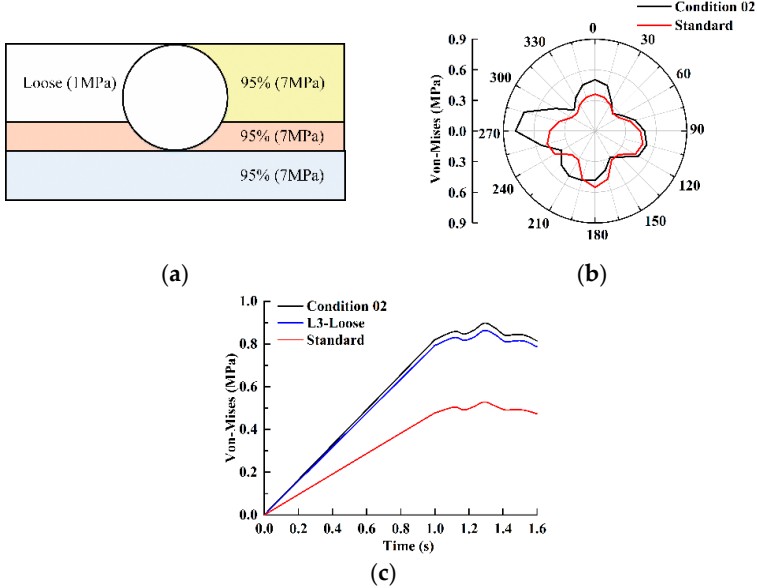

(**a**)　　　　　　　　　　　　　　　　　(**b**)

(**c**)

**Figure 21.** (**a**) The backfilling schematic of Condition 02; (**b**) the stress distribution of the pipe; (**c**) the time history curve of the left springline under Condition 02.

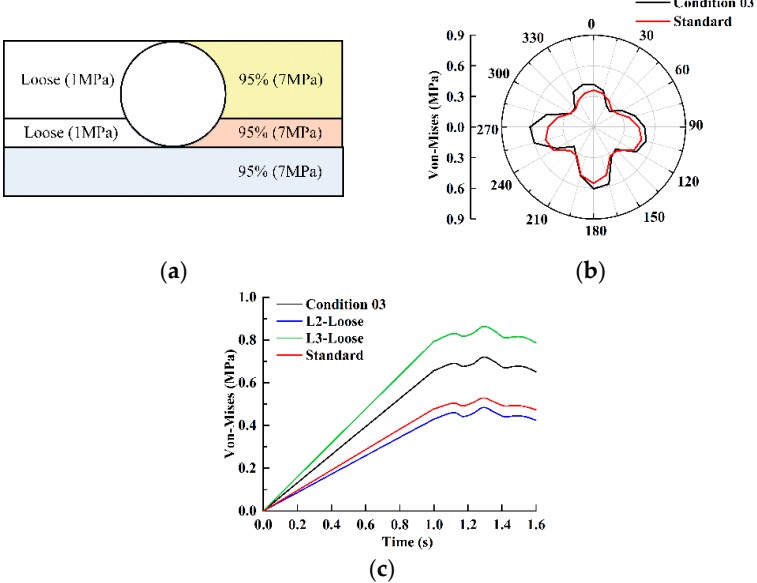

(**a**)　　　　　　　　　　　　　　　　　(**b**)

(**c**)

**Figure 22.** (**a**) The backfilling schematic of Condition 03; (**b**) the stress distribution of the pipe; (**c**) the time history curve of the left springline under Condition 03.

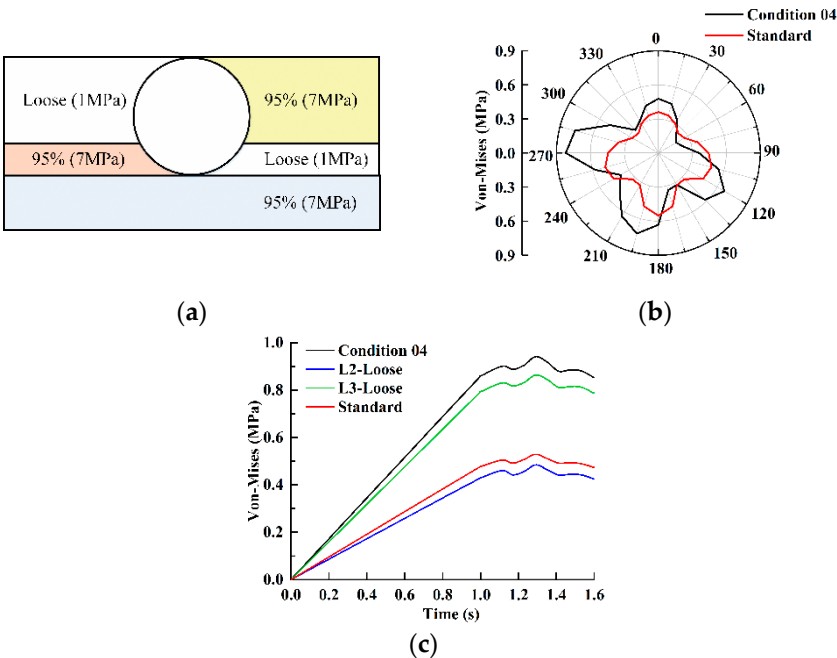

**Figure 23.** (**a**) The backfilling schematic of Condition 04; (**b**) the stress distribution of the pipe; (**c**) the time history curve of the left springline under Condition 04.

## 4. Field Test

### 4.1. General

In the numerical simulation, the double-wall corrugated pipe was replaced by a straight-walled pipe, and the traffic load was implemented by the subroutine. A field test was carried out to verify the accuracy of the dynamic load application method and the 3D model.

For a buried thin-wall flexible pipe, the Von-Mises stress of the pipe is only related to the axial stress and the circumferential stress (Equation (11)) because the shear stress and the stress along the thickness direction are too small. Therefore, the fluctuations in the axial and circumferential stresses of pipes with traffic loads can be adopted to indirectly reflect the variation law of the Von-Mises stress with the traffic load, assuming that the pipe is elastically deformed under the earth pressure and traffic load in the field test. The circumferential strain and axial strain of the pipe were collected, and the Von-Mises stress was calculated using Equation (12).

$$\sigma_{Mises} = \sqrt[2]{\sigma_r^2 + \sigma_x^2 - \sigma_r\sigma_x} \tag{11}$$

$$\sigma_{Mises} = E\sqrt[2]{\varepsilon_r^2 + \varepsilon_x^2 - \varepsilon_r\varepsilon_x} \tag{12}$$

where $\sigma_r$ is the circumferential stress, $\sigma_x$ is the axial stress, $\varepsilon_r$ is the circumferential strain, and $\varepsilon_x$ is the axial strain.

### 4.2. Description of HDPE Pipes and Strain Gauges

An HDPE double-wall corrugated pipe with a length of 12 m, nominal diameter of 600 mm, and a hoop stiffness of 4 kPa was installed in the field test. Strain gauges were arranged in the middle section of the pipe to measure the circumferential strain and axial strain of the pipe at the crown, springline, and invert. Since the interior wall is more susceptible to damage than the exterior wall [62], if the Von-Mises stresses of the simplified straight-walled pipe and the interior wall are consistent, then the numerical model of the straight-walled pipe can be used to predict the deformation and damage of the double-wall corrugated pipe more conservatively. In the field test, the strain gauges

were arranged on the interior wall of the pipe. Four circumferential strain gauges and four axial strain gauges were assembled on the liner, and the same arrangement was applied to the valley (Figure 24). BQ-80AA-P120 electrical resistance strain gauges of 80 × 5 mm by the AVIC Zhonghang Electronic Measuring Instruments Co., Ltd. in China, were used, and the capacity of the strain gauges was 10,000 µε with an accuracy of ±2.2 µε. A DH5921 dynamic stress-strain test and analysis system manufactured by Jiangsu Donghua Testing Technology Co., Ltd. in Jiangsu China, was used to record the measurements, and the system amplified and analyzed the electrical signals collected from the strain gauges and transmitted them to the computer. The system was set to acquire data at a frequency of 50 times per second to ensure the complete recording of strain fluctuations caused by traffic loads. The strain readings were cleared before installation; positive data indicated tensile strain, and negative values indicated compressive strain.

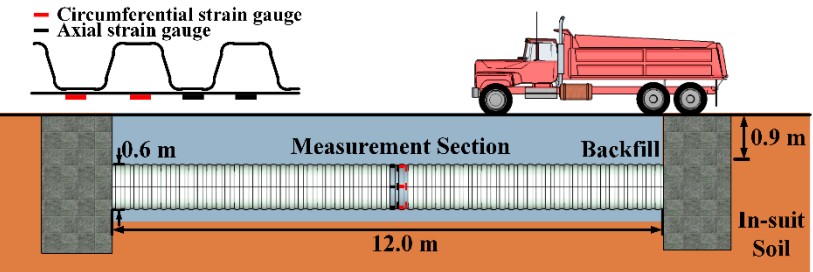

**Figure 24.** Installation of high-density polyethylene (HDPE) pipe and strain gauge arrangement.

### 4.3. Installation and Traffic Load

In the field test, the backfill was medium-coarse sand; the parameters are presented in Table 8. Both ends of the pipe were fixed for the inspection well, and the pipe was installed according to the requirements of the technical specification for buried plastic drainage pipe engineering. A vibratory plate tamper of 450 × 450 mm was used to compact the soil. The stratification and compaction of the backfill during pipe installation are shown in Figure 4. The compaction degree of the backfill was checked by the ring knife method [63], and a truck moving along the axis of the pipe was used to simulate the traffic load. Since the dynamic load factor was 1.2, to ensure that the pressure of each tire applied to the ground was 0.7 MPa, the pressure of each tire acting on the ground when the truck was stationary was 0.58 MPa, and the total load of the truck was 14 t. The wheel distribution and vehicle operating paths are plotted in Figure 25.

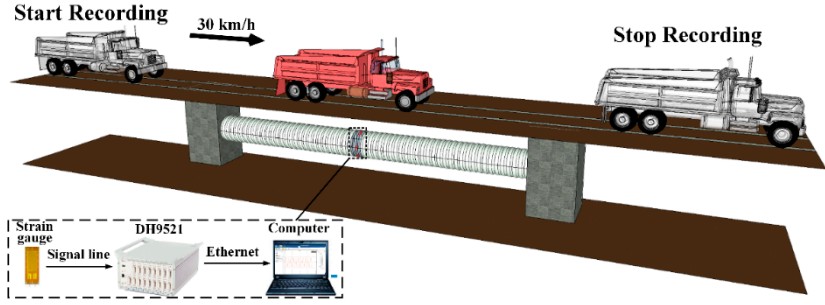

**Figure 25.** Traffic load mode and strain data collection.

**Table 8.** Physical and mechanical properties of the field test soil.

| Properties | Value |
|---|---|
| Existing soil | |
| Maximum dry density (kg/m$^3$) | 1549 |
| Water content (%) | 12 |
| Liquid limit (%) | 25.9 |
| Plastic limit (%) | 17.3 |
| Void ratio | 42.3 |
| Backfill sand | |
| Maximum dry density (kg/m$^3$) | 1730 |
| Coarse sand (0.5–3 mm) (%) | 35 |
| Medium sand (0.35–0.5 mm) (%) | 57 |
| Fine Sand (0.25–0.35 mm) (%) | 8 |
| Grade | 120 |

## *4.4. Comparison of Test Results*

The variation in the interior wall strain with time under the traffic load is shown in Figure 26. The Von-Mises stress fluctuations in the valley and liner were roughly calculated according to Equation (12), and they are plotted in Figure 27. The condition of the field test was simulated using the above 3D model, and the Von-Mises stress of the pipe was calculated. The measured data were compared with the simulation results, which showed that the straight-walled pipe could effectively simulate the Von-Mises stress of the interior wall at the crown and the invert. At the springline, the Von-Mises stress of the straight-walled pipe was close to that of the liner but much smaller than that of the valley. This might be explained by the field test method: the traffic load was applied on the soil surface on the 6th day after the installation was completed, and the strain of the pipe was recorded. At this time, the deformation of the pipe and the soil had not yet reached a stable state, making the resulting measurement larger. However, regardless of whether the liner or valley was measured, the stress fluctuations caused by traffic loads were very similar, which proved that the method used for applying the traffic load by the subroutine in the numerical simulation was feasible. In sum, there was an error in the method for predicting the deformation of the interior wall by the Von-Mises stress of the straight-walled pipe, but the changing trend was similar. The feasibility of the three-dimensional numerical model used in this study was verified.

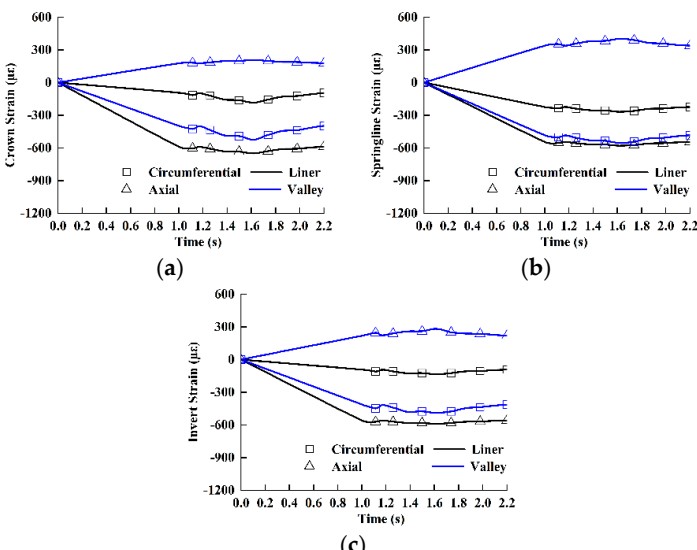

**Figure 26.** The measurements of pipe strain under traffic load: (**a**) the crown; (**b**) the springline; (**c**) invert.

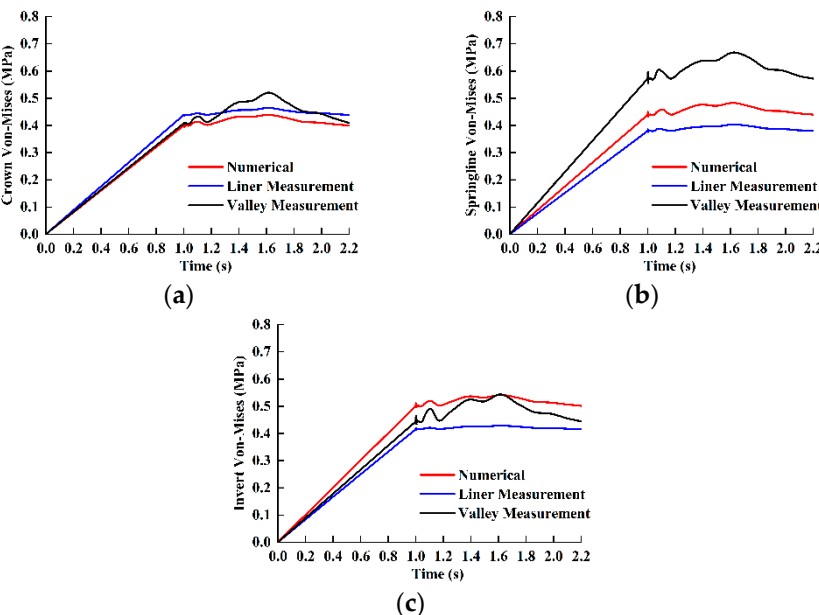

**Figure 27.** Comparison of measurement and numerical simulation of pipe Von-Mises stress under traffic load: (**a**) the crown; (**b**) the springline; (**c**) invert.

## 5. Conclusions

In this study, a double-walled corrugated pipe was simplified into a straight-walled pipe on the basis of hoop stiffness, and a three-dimensional finite element model was established. The changes in the Von-Mises stress of the pipe under a real traffic load and loose backfill were analyzed. For the tested conditions, the following conclusions could be drawn.

1.  In the numerical simulation, the Von-Mises stress of the straight-walled pipe can be used to predict the deformation response of the interior wall of the corrugated pipe under an external load, and the method for simulating the real traffic load by combining the Vdload subroutine with the half sinusoid is credible.
2.  In the backfilling process, the maximum Von-Mises stress of the pipe is several times that of the stable state, the time during which the pipe is in this high-stress state is very short, and the influence on the pipe is very limited. The Von-Mises stress of the pipe is mainly determined by the earth's pressure at the crown. The increase in the buried depth of the pipe increases the deformation of the pipe and reduces the influence of the traffic load on the pipe.
3.  Reducing the compaction degree of the backfill in contact with the pipe causes greater deformation of the pipe. Loose L1 has a negative effect on the pipe; the decrease in the compactness of L2 leads to an increase in the invert stress and a decrease in the stresses of the crown and the springline, and a decrease in the compactness of L3 results in an increase in stresses at the crown and the springline and decreases in the stress at the invert.
4.  Compared with the earth pressure, the impact of the traffic load on the pipe is very limited. Pipes are more sensitive to traffic loads when the compaction degree of the backfill in contact with the pipe decreases. The response of the Von-Mises stress of the pipe to traffic loads is determined by the magnitude and the speed of the load, and the maximum Von-Mises stress is proportional to the traffic load magnitude and independent of the load speed.
5.  The diameter and the hoop stiffness of the pipe affect the response of the pipe to the compactness of the backfill and the traffic load. The compactness of the backfill soil around the pipe needs to be higher when the diameter of the pipe is small; the variation in the hoop stiffness has a limited effect on the Von-Mises stress of the pipe, and there is no obvious law.

6. If the compactness of the backfill on the left and right sides of the pipe is not consistent, then the Von-Mises stress concentrates near the region of the pipe in contact with the loose backfill. The Von-Mises stress fluctuation caused by the traffic load is more significant than the stress under standard conditions, but it can still be negligible.

**Author Contributions:** Conceptualization, H.F.; Data curation, P.T., B.L., K.Y., and Y.Z.; Funding acquisition, H.F.; Investigation, P.T., B.L., K.Y., and Y.Z.; Methodology, P.T.; Software, P.T.; Writing—original draft, P.T.; Writing—review and editing, X.D. All authors have read and agreed to the published version of the manuscript.

**Funding:** This research was funded by the National Key Research and Development Program of China (No. 2016YFC0802400), the National Natural Science Foundation of China (No. 51678536), the Scientific and Technological Research Program of Henan Province (No. 152102310066), and the Outstanding Young Talent Research Fund of Zhengzhou University (1621323001), for which the authors are grateful.

**Acknowledgments:** The authors would like to thank the anonymous reviewers for their constructive suggestions to improve the quality of the paper. We thank MDPI for its linguistic assistance during the preparation of this manuscript.

**Conflicts of Interest:** The authors declare no conflict of interest.

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
