# Peer review of "Numerical and Experimental Investigation of the Effect of Traffic Load on the Mechanical Characteristics of HDPE Double-Wall Corrugated Pipe"

_applsci, doi:10.3390/app10020627_

Round 1
Reviewer 1 Report
General
The subject of the manuscript is very interesting and of practical relevance in the field of Civil Engineering and particularly in the areas of soil mechanics and hydraulics. The manuscript describes a very large volume of work of simulations performed using the well-known ABACUS commercial software package but also some field measurements with the purpose of validation of some numerical results. The work seems generally well planned and well conducted but the current version of the manuscript is full of errors of all type, language deficiencies and lack of accuracy in the presentation of the scientific contents. These issues may cause serious doubts in the readers about the validity of the work. Several results and conclusions, in particular, are presented in a way too much generalist or are badly explained, ignoring the particularities, conditions, or assumptions for which they were obtained. In the reviewer understanding, the entire manuscript can/must be significantly improved and requires a profound and major revision, in language, format or style, and also is some details of the contents. The aid of an expert in the field of soil mechanics, or of a native English speaker, and/or of a Professional Editing Service would be recommendable.
The influence of pipe cover depth, pavement design, and compactness of the backfill above the pipe that are fundamental on the static or dynamic loads that are applied to the buried pipes are never analyzed nor discussed in the current version. Their characteristics are very important in the degradation of the static or dynamic loads applied at the surface. Ignoring these fundamental issues may lead to results, comparisons or conclusions that may be only partial and may be biased or out of reality. An example of inexactness in Line 15 of the abstract after “networks: Instead of ”Backfill compactness and traffic load are the major factors”, it should be “Characteristics of surrounding soil and pipe bed, pipe cover depth, backfill compactness, type of pavement and pavement design, and traffic load are some of the major factors”, or similar. Also the term backfill is used vaguely along all manuscript without specifying exactly what part of the trench it is intended to mean, normally the lateral backfill of the open trench or even the pipe bed in the current version. Also, this manuscript seems to deal only with construction in open trench, excluding other situations, as the cases of construction in pure backfill, and tunneling.
Specific
According to what was written above, many details need to be corrected or improved, including language issues. Below are mentioned some of the more relevant.
Lines 16 to 19 – Please consider “In this paper, the ABAQUS 3-D finite element model was used for the study of the influence of backfill compactness, traffic loads, diameter, and hoop stiffness on mechanics characteristic for HDPE pipe under the traffic load. A series”, or similar, instead of “In this paper, a 3-D finite element model for the study of the influence of backfill compactness, traffic loads, diameter, and hoop stiffness on mechanics characteristic for HDPE pipe under the traffic load, was established base on ABAQUS. And a series”.
Line 19 – Please consider “the validity of the simulation results.”, or similar, instead of “the validity of the model.”
Line 20 - Please consider “1) traffic load alone has little influence on the mechanical behavior”, or similar, instead of “1)only under traffic load, it has little influence on the mechanical characteristics”
Line 24 - Please consider “non-compacted backfill easily cause”, or similar, instead of “uncompacted backfill is easy to cause”
Figure 1 is mentioned in page 1, but only is presented in page 3.
Lines 42, 45, 61, 73 and along all introduction. The year of the publication could be given after the name of the authors: “Talesnick et al. (2011)”, instead of “Talesnick et al.”; “Terzi et al. (2012)”, instead of “Terzi et al.”; “Abri and Mohamedzein (2010)”, instead of “Abri and Mohamedzein”; “Mohamedzein and Aghbari (2016)” instead of “Mohamedzein and Aghbari” …
Lines 58 to 60 – This text is confusing and perhaps should be rewritten. Lines 64- 65 – What is the intended meaning of “normal fault”. Also this sentence seems grammatically incorrect.
Line 70- What is “pipe invalidation”? Please correct or explain better in the text.
Line 85- What is “low-altitude traffic load tests”? Any relation with “low pipe cover depth”? Please correct or explain better.
Lines 91-92 – Please consider “However, there is little information about the mechanical response”, or similar, instead of “However, there is no relevant research on the mechanical response”.
Last Paragraph of the Introduction after line 92 – Please revise and/or explain better. Line 95 – Perhaps “based” instead of “base”.
Line 119 – What is exactly “the normal diameter of the pipe”?
Line 122 – How is “the wall thickness of the straight-wall pipe” exactly calculated? Perhaps the wall thickness “t” could also be represented in Figure 2.
Line 134 - Subsection 2.2 – Please consider “2.2. Data of the road structure”, “2.2. Dimensions of the road structure”, or similar, instead of “2.2. Model of the road structure”.
Line 148 – Please consider “Figure 4” perhaps instead of “Figure 3”; “numerical simulation”, instead of “numerical analysis”.
Line 149 – Please consider “Figure 3” perhaps instead of “Figure 4”; “data”, instead of “model”.
Line 150 – Please consider “Table 3” instead of “Table 1”; “numerical simulation”, instead of “numerical analysis”.
Line 151 – Please consider “Table 4” instead of “Table 2”; “numerical simulation”, instead of “numerical analysis”.
In Tables 3 and 4 “Degree” intends to refer to “Density”? The order how Figures and Tables were mentioned in the text of this subsection were not coherent with the order of their presentation. Also the order was very confusing. As proposed above, current Figures 3 and 4 could change their numeration (and order of presentation), and the order could be then now: Figure 3 (old 4), Table 3, Figure 4 (old 3), and Table 4. A possible solution, or similar (last sentence of this subsection is now the first sentence):
“2.2. Data of the road structure
The whole 3-D data of the road structure and the traffic load location is displayed in Figure 3. To reduce the influence of the boundary on the pipe, a model with dimensions of 16 m×20 m×6m is adopted [1]. The dynamic load is applied to the region (0.3m×10 m) of the wheel track with the wheelbase of 1.7 m.” … Lines 141-142 - “are shown in Figure 4”, instead of “are shown in Figure 3.”… The subsection ends now with (Line 145) “planes are not allowed [30].” Please confirm and revise accordingly or similarly.
Line 152 - Please consider “2.3. Data of the traffic load”, or similar, instead of “2.3. Model of the traffic load”.
Lines 153 to 155 – The type of vehicle considered (a 6 (4+2) wheel truck? as it seems to be in Figures 21 and 22) and the loads transmitted should be better described in this subsection. With this regard, Figure 5 could be improved.
Line 162 - Please consider “point is given by”, or similar, instead of “point was shown in”.
Line 167 - Please consider “is”, instead of “was” (twice in this line).
Line 174 - Please consider “a fixed time”, or similar, instead of “the fixed time”.
Lines 174 to 176 – “iterative calculation”, “iteration completes” “after two iterations” - No iterative procedure is exactly described here.
Line 183 and followings – Beginning of subsection 2.4. – It could be explained why was not analyzed or discussed the compactness of the backfill or layers above the pipe.
Line 200 - “Table 5”, instead of “Table 3”. This Table can be improved. “(MPa)” Could be removed from all lines of the left side and introduced in an upper line above “Compaction Degree (%)” in the right side of the Table as “Elastic Modulus E (MPa)”, or similar.
Line 201 - Table 6 is very confusing and can/must be improved. It is not clear how the left and right parts of the Table combine. Some text between lines 202 and 207, at least, does not help much, and seems even unnecessary. It seems that seven options for the backfill conditions in the pipe bed and low part of the trench were tested. The number of all combinations tested perhaps could be better explained in the text, and the Table could be improved. In the end part of the left side of Table 6, where it is “L2” (twice), it seems that should be “L3”. Perhaps Table 6 could be split in two parts 6a) and 6b). Part “6a) Backfill conditions” could be improved with 4 columns, instead of one column, describing the 7 backfill options and the compactness of L1, L2 and L3, with the inclusion in the Table 6a) of what the standard conditions exactly are (first line). Text between Lines 202 to 207 could be then removed. Other alternatives are naturally possible. Please confirm and revise.
Lines 208 to 224 and lines 386 to 400 – Between a section and a subsection there should be no text. A possible solution, maintaining generally all text and its current position, is to introduce two subsection titles immediately after lines 208 and 386 respectively “3.1.General” and “4.1. General”, or similar. All numeration of the subsequent subsections should then be increased by one: Lines 224, 247, 273 …“3.2, 3.3, 3.4 …” instead of “3.1, 3.2, 3.3 …”; Lines 400, 419, 433, “4.2, 4.3, 4.4” instead of “4.1, 4.2, 4.3”.
Line 220 – Perhaps “in the following subsections”, or similar, instead of “in the next part”
Line 223 – With “of the middle section of the pipe” it is intended to mean “in the section located at half-length in the pipe longitudinal direction”, or similar? It should/could be better explained what is the meaning intended by the term “Mises stress” that is a key word and is repeatedly mentioned along the manuscript and how is calculated. According to Mises Criterion it would be a stress in the limit of the elastic state. The stresses presented in all Figures seem all inside the elastic state and very far from that limit that is never given in the text nor represented in the Figures.
Line 225 - Please consider “In these simulations, the magnitude and speed of the traffic load are set as 0.7 MPa and 60 km/h, respectively, as base criterion.”, or similar, instead of “In this part, the magnitude and speed of the traffic load are 0.7 MPa and 60 km/h, respectively.”
Lines 230-231 – Here and along all text. The backfill in analysis should be specified. It should be noted that, according to the current version, the compactness of the backfill above the pipe (L4, L5, and L6) was always constant in the simulations. Please consider “When the compaction degree of the pipe bed and lateral backfill decreases”, or similar, instead of “When the compaction degree of the backfill decreases”.
Line 240 -241 – The unfavorable effect that the deficient backfill compaction and deficient pipe bed conditions can have on the flexible pipe deformation or flexion, differential settlements, failures in the pipe joints, and consequent bad hydraulic performance is not mentioned here nor in any part of the text in the current version. These aspects are fundamental either in the design phase as in the operating conditions of any sewer network and should be highlighted. A network in bad performance conditions may increase greatly the infiltration from the water table and the operating costs, particularly in the wastewater treatment, and may require rehabilitation for reducing infiltration. With this regard see “An effective and comprehensive model for optimal rehabilitation of separate sanitary sewer systems”, AF Diogo, LT Barros, J Santos, JS Temido, Science of the Total Environment 612, 1042-1057.
Figure 7 can be improved. The “specified value” or limiting value for the stress corresponding to the limit of the elastic state of the pipes is not shown in Figure 7a) b) and c), reviewer supposes. Vertical axes represent a stress or a tension (MPa). Legend of vertical axis can be improved. Please consider “Computed Stress (MPa)”, “Mises Stress (MPa)”, or similar, instead of “Crown Mises”, “Springline Mises”, and “Invert Mises”. The same for Figures 10 to16 and 24.
Caption of Figures 8-9-10-11–12-14-15-16: Please consider “(a) Change in L1 compaction degree; (b) Change in L2 compaction degree; (c) Change in L3 compaction degree.”, or similar, instead of “(a) uncompacted L1; (b) uncompacted L2; (c) uncompacted L3.”
Figures 8 to 12 and 14 to 16 are practically unreadable in the copy that reviewer received. Please improve, if possible.
Line 289 – Perhaps “The variation”, instead of “The varies”?
Lines 292 to 297 and in other instances – The laws of variation of Mises stress with the parameters/characteristics tested in general (and with the diameter in this case), and the corresponding proportionalities do not seem to have been fully obtained or demonstrated in this work. Please revise when applicable.
Line 325 – Perhaps “a) the crown”, instead of “a) the invert”. Please confirm and correct.
Line 355 – Perhaps “are analyzed”, or similar, instead of “are designed”.
Equation 8, and Lines 397 to 399 - Notation - Perhaps “σxx, σyy, σzz”, or similar, instead of “σx, σy, σz”? (Coherence with the last parcels inside the square root). It is said that “σr” is the “axial stress” in line 398. Perhaps “σx”, or similar, instead of “σr”? The indices “r” is normally more associated with the radius than with axis. Please confirm and correct, if that is the case.
Line 401 – It is written “pipe with a length of 12 m”. In Figure 21 the pipe length seems to be 10 m. Please confirm and correct for consistency, if that is the case.
Figure 21 and Figure 3 – A pipe cover depth of 0.9 m is presented, but this value is never explained or discussed. Minimum pipe cover? Also due to hydraulic requirements, pipe cover depth may be not always constant in the longitudinal direction of the pipe. This was never discussed.
Line 436 – Perhaps “data”, or similar instead of “model”.
Line 437 – Perhaps “results”, or similar instead of “data”.
Line 444 – Perhaps “utilization”, or similar instead of “established”.
Conclusions Section should be deeply and consistently revised and corrected accordingly with all above comments, corrections, and recommendations.
Reviewer 2 Report
The paper deals with a topic of some importance: the deformation of double-wall corrugated pipe under traffic load. It is of importance, since this construction is used quite often to underpass roads or railways.
The English of the paper is below standard. I started with some remarks, but since I am a reviewer and no editor, nor a native speaker, I stopped with that. The paper can only be accepted when edited by a native speaker.
A problem I have with the approach that the pipe is ‘wished in place’. There is a pile in densified soil and then the stresses caused by increased loading (traffic) are calculated. In reality the pile has to be installed and the soil has to be densified. This densification causes already a large deformation in de pipe and the traffic loading is additional deformation. Therefore, the loading on the pile is larger in reality than simulated. The consequence of this has to be discussed in the paper.
The paper deals only with the results of the numerical calculation. The paper would be much more valuable when the results are coupled with the results of analytical analyses. It seams that the soil is stiff compared to the pile. In that case the pipe will deform in more or less the same way when it is a bit stiffer or less stiff, independent of the stiffness, because the soil densification determines the deformation of the pipe. This can be shown (or not) by some plots of vertical deformation. It is strongly advised to rework the paper and to include a comparison with analytical methods.

Reviewer 3 Report
Comments & suggestions are shown in attached file.

Round 2
Reviewer 1 Report
The manuscript presentation has improved greatly in this new version. Reviewer sincerely congratulates the authors for their good work and for the kind words in the authors responses to Reviewer. According to the authors, most of the recommendations of the first comprehensive, detailed and constructive review report performed by the Reviewer were fully followed. However minor revisions seem still required. Please check again first detailed revision report performed by the Reviewer with respect to any eventual details that were not fully considered or totally corrected in the text of this new version.
Some few examples of possible minor corrections, between others:
Line 21 – Perhaps “For the conditions tested, the results showed the following”, or similar, instead of “The results showed the following”. As clearly pointed in the first revision of the Reviewer, it should be noted that the pavement design and pipe cover depth, at least, have a vital role in the surface loads degradation.
Lines 456-457 – Perhaps “were analysed. For the tested conditions, the following conclusions can be drawn.”, or similar, instead of “were analyzed, and the following conclusions can be drawn.”
Lines 137-138 – The sentence “The whole 3D data of the road structure and the traffic load location are displayed in Figure 3.” is repeated in these lines unnecessarily and perhaps should be removed.
Line 138 – Consider perhaps “The thickness”, or similar, instead of “In addition, the thickness”
Line 138 – Consider perhaps “considered in the numerical simulations”, or similar, instead of “in the numerical analysis”
Line 144 - Consider perhaps “Table 3 … in numerical simulations”, or similar, instead of “Table 1 … in numerical analysis”
Liner 145 - Consider perhaps “Table 4 … in numerical simulations”, or similar, instead of “Table 2 … in numerical analysis”
According to the current text version, Table 3 and Table 4 should be inserted between current Figures 3 and 4. Tables and Figures should be presented in the order that they are mentioned in the text.
Line 204 – Perhaps “Table 5”, instead of “Table 3”.
Line 205, and following pages – Perhaps presentation/aspect of current Table 6 and also some (several) Figures (if possible) can still be improved.
Reviewer 2 Report
As in the previous version, my greatest concern is that the pipe is ‘wished in place’ and the influence of compaction of the various soil layers on the deformation of the pipe is not taken into account. Therefore, the influence of the traffic loads may be calculated well, but the deformation that was there before the traffic load is not known.
This I have mentioned in the first version but has not been changed. It is now to the editor, if this has to be changed for acceptation or not. My opinion that the deformation during compaction of the soil layers is of paramount importance and has to be included before the paper can be accepted.

Round 3
Reviewer 2 Report
Now the influence of installation and how the density is measured is covered.
There still were some typos in the text. The one I have seen are marked in hte text, but I assume an professional editor will find more.
